# Data-Driven Threat Analysis for Ensuring Security in Cloud Enabled Systems

**DOI:** 10.3390/s22155726

**Published:** 2022-07-30

**Authors:** Mohammed K. S. Alwaheidi, Shareeful Islam

**Affiliations:** 1School of Architecture Computing and Engineering, University of East London, London E16 2RD, UK; u1724270@uel.ac.uk; 2School of Computing and Information Science, Anglia Ruskin University, Cambridge CB1 1PT, UK

**Keywords:** threat modelling, data level, cloud based system, data flow diagram, control, cloud service provider

## Abstract

Cloud computing offers many benefits including business flexibility, scalability and cost savings but despite these benefits, there exist threats that require adequate attention for secure service delivery. Threats in a cloud-based system need to be considered from a holistic perspective that accounts for data, application, infrastructure and service, which can pose potential risks. Data certainly plays a critical role within the whole ecosystem and organisations should take account of and protect data from any potential threats. Due to the variation of data types, status, and location, understanding the potential security concerns in cloud-based infrastructures is more complex than in a traditional system. The existing threat modeling approaches lack the ability to analyse and prioritise data-related threats. The main contribution of the paper is a novel data-driven threat analysis (d-TM) approach for the cloud-based systems. The main motivation of d-TM is the integration of data from three levels of abstractions, i.e., management, control, and business and three phases, i.e., storage, process and transmittance, within each level. The d-TM provides a systematic flow of attack surface analysis from the user agent to the cloud service provider based on the threat layers in cloud computing. Finally, a cloud-based use case scenario was used to demonstrate the applicability of the proposed approach. The result shows that d-TM revealed four critical threats out of the seven threats based on the identified assets. The threats targeted management and business data in general, while targeting data in process and transit more specifically.

## 1. Introduction

Cloud computing offers many benefits in terms of its wider adoption including cost savings, less maintenance overhead, and on-demand capacity growth. The overall market of cloud computing is predicted to increase at a 16.3 percent annual pace from USD 445.3 billion in 2021 to USD 947.3 billion by 2026 [1]. Regardless of its wider adoption, it is critical to understand the security concerns that businesses may encounter when considering cloud-enabled systems for overall business continuity. Threats and vulnerabilities relating to the cloud services (i.e., Infrastructure as a Service, Platform as a Service, Software as a Service, or Container as a Service) [2] can pose potential risks. According to the Cloud Security Alliance, the top three threats in cloud computing include data breaches, misconfiguration, and insufficient change control, as well as a lack of cloud security architecture [3]. Threats analyses for cloud -based systems are increasingly becoming essential for all systems. There are a number of threat analysis techniques but a lack of focus on the comprehensive understanding of data for threat analysis is apparent.

This paper presents a novel threat modelling approach that initiates from understanding business processes, services and infrastructure. Infrastructure is categorised into five attack layers by considering data from the user agent to data storage. The model classifies the data and uses a DFD diagram, which aims to gather common weaknesses and threats so that appropriate control actions can be taken into consideration to tackle the threats. The d-TM classifies data in order to distinguish between the various risks associated with each type as well as to determine which data are more susceptible to threat. There are four main contributions of this work. Firstly, the d-TM describes data abstraction in terms of three levels, i.e., control, management, business and phases. Each data level has its own set of security concerns that require adequate protection. The data levels are divided into the following three phases: data at rest, data in transit, and data in the process. With such a data classification, an organization gains greater control over any data stored, processed, and transported from and to the cloud system. Secondly, d-TM provides a systematic process by which to understand organizational data and services and data flow for threat analysis and control. The identified threats are prioritised based on a number of factors including business-as-target, threat complexity, and overall impact. Thirdly, it adopts the widely used security knowledge from existing standards including Common Attack Pattern Enumeration and Classification (CAPEC), Common Weakness Enumeration (CWE) and NIST SP800-53 for threat analysis and control. It provides a wider adoption of d-TM for the cloud-based system context. Finally, a real cloud-based use case scenario was used to evaluate the applicability of the d-TM. The results show that d-TM can analyse and prioritise threats based on an understanding of the data from business processes and services. The data flow model also provides a common understanding of the data levels and phases from internal and cloud infrastructure so that weaknesses and associated threats at each level can be analysed to determine suitable control for overall security assurance.

## 2. Related Works

There are several existing works that focus on threat analysis and threats relating to the cloud. This section provides an overview of the existing works that are relevant to our approach.

### 2.1. Threat Analysis Model and Standards

Cloud-based systems, similarly to traditional data servers, present threats that, if not managed properly, can impact data integrity and security. The first model is a collection of threats known collectively as STRIDE [4], which stands for Spoofing identity, Tampering with data, Repudiation threats, Information disclosure, Denial of Service, and Elevation of privileges. The significance of the STRIDE technique in studying the security of cloud systems has been emphasised in investigations on the resilience of cloud systems. According to Abdulsalam and Hedabou, the STRIDE technique is an effective measure for safeguarding the cloud system from threats [5]. The authors claimed that STRIDE helps to demonstrate an attack vector’s impact before its occurrence in cloud systems. The STRIDE approach combines different security threats with strategies that can help cloud service providers understand a cloud system’s vulnerability. PASTA is another potential threat modelling system that stands for Process for Attack Simulation and Threat Analysis. The technique of conducting PASTA in cloud systems involves seven stages, each building on the previous until the cyber security posture of the cloud system is fully assessed. Thus, this method is effective because it examines the intricate cloud model components that could complicate the business with threats [6]. The attack tree provides a schematic representation of how assets in a cloud system might be attacked in the form of a tree. This threat analysis system is an irrefutable must-have for organizations operating on cloud systems because it presents the most vulnerable assets [6]. An attack tree of a man-in-the-middle cloud attack is a modelled example of the way a cloud system on synchronization can survive an attack directed towards an application or specific data. Therefore, an attack tree determines the most vulnerable cloud resources.

The NIST SP 800-154 [7] focuses on data modelling and security surveillance. The SP 80-154 guideline eliminates the conventional best practices by allowing organisations to use it as a central data model for risk management [8]. Despite the fact that the model has been in draft form since 2016, it is an alternative for cloud computing-based organizations since it helps businesses to avoid risks by taking proactive efforts to limit threats to data before they arise. The IEEE established the Cloud Service Provider (CSP) Standardization Program to guarantee that cloud computing suppliers satisfy specific criteria that reduce risk and security issues associated with cloud computing services. This standard specifies a broadly shared function model for cloud computing, with the goal of standardizing how functions are shared between cloud service providers (CSPs) and cloud service clients (CSCs) [9].

### 2.2. Cloud Threats

Cloud threats exist, as they do for any other digital technology. However, security is critical to the widespread use of cloud computing services [9]. The findings of [2] revealed seven key security threats to cloud computing services, including data manipulation and leaking, which were among the most common threats. Other security threats related to data infiltration and data storage in the cloud computing environment were found. A review of cloud migration goals including cost savings, collaboration and sharing, scalability and IT efficiency and risks such as data leakage, service unavailability, lack of trust and controls were discussed in [10]. The work emphasised the necessity of a comprehensive risk management framework and the monitoring of evolving risks and threats in the cloud-based context. A survey identifies a list of threats in the cloud, notably data lock-in, performance unpredictability, abuse of cloud resources and data breaches which are grouped into threats to the application, data and infrastructure [11]. The threat model proposed by [12] aims to analyse the threats in cloud data infrastructure. Two different cloud infrastructures were considered for the threat analysis based on an attack surface with regard to users and cloud providers and attack trees and graphs were used for the threat analysis. An ontology-based approach was considered for the threat analysis by [13], where ontology was used to model the actors and their relationships and threats were considered from the design structures matrix of various actor perspectives. Mitsis, G. et al. proposed an optimised framework for Mobile Edge Computing (MEC) server selection by the end users [14]. The server selection follows the reinforcement learning technique which adopts the theory of the stochastic learning automata. The result shows that the approach provides a better offloading performance. A threat intelligence approach was proposed to analyse the threat within a cloud honeypot as a service by applying elastic search [15]. Several parameters such as Matrix IoC, attack, behaviour and pattern were used for the threat analysis. The experimental results identified a significant number of attacks in the cloud honey pot as a service. A risk assessment method for a cloud-based system was proposed using a cloud risk matrix [16]. In particular, frequency, severity, and risk index were the three variables used in the risk matrix which were applied in three critical procedures including fuzzification and randomization, inference process and defuzzification. The results from the case study show the effectiveness and rationality of the proposed method. A review of the existing risk assessment methods and their suitability for the cloud-based system is provided in [17]. The work proposed a new risk assessment model (CSCCRA) that considers Cloud Quantitative Risk Analysis (CQRA), Cloud Supplier Security Assessment (CSSA), and Cloud Supply Chain Mapping (CSCM). The approach was compared with the three other risk assessment methods and it was observed that CSCCRA was the superior method.

To summarise, the works reviewed above provide feasible approaches by which to understand and analyse cloud risks in domains where traditional threat modelling methodology does not place a strong emphasis on data. The majority of threat analyses focus on topics such as assets, strategies, and threat modelling. Furthermore, standards such as NIST SP 800-154 concentrate on business data for systems that lack a thorough understanding of all data from the entire cloud system environment, such as control data. Our study solved these issues by providing a native data-driven threat analysis model incorporating the full technological coverage of numerous types of data in the cloud at any point in its lifetime. The proposed approach also considered business processes and services as the initial reference point to analyse threats that are overlooked by existing studies.

## 3. Abstraction of Data Level

Data in the cloud are different to in traditional infrastructure due to transmission and storage to the Cloud Service Provider (CSP) infrastructure. The d-TM recognizes the importance of understanding various types of data throughout its lifecycle. d-TM classifies the data into the following three abstracted levels: management, control, and business and each level considers three states of the data, i.e., at rest, in process, or in transit.

Management Data-level (mD) is data generated by the organisation related to managing the cloud components such as—the cloud admin portal or cloud computing resource. The data could include authentication or access methodology.Control Data-level (cD) is related to any data that support technology functions such as Application programmable interface (API) calls, routing information, application inquiries, configuration updates, logs, action events or data backup and replication. Control data can be purely operational data or something more to businesses, such as an API request for customer information or identity validation. However, data seeks to be communicated between local and cloud systems.Business Data-level (bD) refers to any data related to business services that business users generate to access organisation cloud services such as emails.

Businesses should protect data that might expose any potential risk, especially when business and operational data are inextricably linked. For example, a weakness in API communication between cloud components might result in data leakage. In addition, account takeover attacks might have a significant impact due to insecure cloud services. Apart from the control and management-level data, the cloud platform also requires business-level data. This data runs on the system, and it entails information between cloud service users and CSP. This data needs to be secured in the cloud platform for organisations using a cloud system, although it remains vulnerable to attacks. Depending on the level of security at each management and control level, this data can be secured or exposed to the public [18].

Data is considered at Rest (Dr) when it is stored in CSP infrastructure or in a remote server [19]. Moreover, the data reaches the final stages of the cycles in the cloud, where it should be saved for long-term uses [20]. It can be attacked, modified, and erased from the system while still at a distant place under the control of system security.Data in Process (Dp) refers to data at rest that have been accessed by a user or service for a brief period of time, which may be used during the execution of an application. Although such data are not inherently suspicious, a malicious service, software, or hardware might change or leak it. A security breach may be signaled if data are obtained by unauthorised access to system resources via Advanced Persistence Threat or malicious hardware components. For example, an Amazon AWS data leak attack was caused by a rogue Microchip on Supermicro servers utilised by AWS [20].Data in Transit (Dt) refers to when data are shared with other systems or users. Data in transit are particularly vulnerable to cloud computing systems, especially when moving through unsecured networks or to an application programming interface (API) that lets programs connect with one another. Business data and operation data, such as admin management identity, are transmitted via the internet in a cloud environment. Data in transit are vulnerable to a variety of cyberattacks, including Man-In-The-Middle (MITM) attacks [21]. According to [22], regardless of the increased regulatory focus on data security, this data phase should be protected since a data breach can influence multiple levels of a business, such as sensitive data exposure, which can result in reputational harm or financial penalties.

Figure 1 shows the data levels and phases at any cloud infrastructure. The figure depicts data initiated by any cloud user to three abstraction levels. Each level is related to the user type (actor). The actor may be a business user, an operator, or another type of system. Each data level could be at any phase of storage, processing, or transmission. However, data phases are linked to cloud infrastructure, which is made up of the following three main components: CPU, memory, and input/output function (network).

## 4. The Data Driven Threat Analysis

The section presents the proposed d-TM that provides a structured way to identify and analyse threats from the overall cloud-based ecosystem. The proposed data-driven threat model (d-TM) emphasizes a holistic consideration of threats from all phases of data so that appropriate control actions can be determined for overall security assurance. It is based on business operations and technological logic, and investigates the organisational data. The research methodology used in this research was initiated using d-TM concepts that include threat layers, actors, and a common knowledgebase, followed by a threat analysis process, and finally the method was evaluated using a case study scenario.

### 4.1. Justification for Using Data-Driven Approach

Data represent the most valuable asset for any business; therefore, it is critical that threats are assessed from all aspects of the data. Today, the majority of threat analyses techniques attempt to reduce system risk to a minimum when an examination of the assets, systems or threat actors is insufficient—especially when the business is considering moving to the cloud. The cloud idea is centred on data. Hence, when an organisation migrates to the cloud, it mainly shares its data with the cloud provider. This is why a data-driven strategy that empowers the d-TM approach is critical for cloud-based solutions.

### 4.2. d-TM Deployed Concepts

This research considers a number of concepts for the data-driven threat analysis which are presented below.

#### 4.2.1. Threat Layers in Cloud Computing

The threat analysis approach is offered based on the generalised model of cloud architecture. Cloud plans are continuously evolving to provide greater flexibility and cost savings. Therefore, threats and weaknesses differ depending on the type of the chosen cloud model. The threat analysis technique should be flexible enough to handle various cloud types. As a result, the proposed technique was built on a tier-based approach, allowing the organisation to examine threats to data at any point of the infrastructure regardless of data location. The model takes into account that each layer should be easily linked to the IT roles and response capabilities, such as the application layer—developers, computer layer—system administrators, network layer—network team, and so on. d-TM considers five layers that could provide a possible attack surface. Layers present the path of any data flow from user to cloud and vice versa, and layers are interconnected at points where businesses could be disrupted if any layer is not built to be secured.

The first layer is the **Agent** that provides insight into the required tool for cloud users to access data or services such as web browsers. The web browser is used to access any cloud services and a compromised or vulnerable browser could lead to severe threats to data. The browser could hold sensitive information, i.e., session details, encryption keys, or saved credentials. This information could be compromised while saved or processed by rouge software/plugin [23] and also during transition by network attacks such as MITM [24]. It is important to keep the user agent secured because the user could be a cloud admin who accesses the cloud portal for day-to-day operations, and the data could be compromised to a such a level that could lead to cloud takeover due to stolen admin credentials.The second layer is the **Network** which aims to identify any device that interacts with data prior to reaching the business cloud service or server. This layer could include physical or virtual routers, switches, firewalls, or load balancers. Business data for instance at this stage are in a transit state and could be impacted by any configuration manipulation that leads to data leakage. This kind of attack is not limited to the attacker only, as the cloud admin could disrupt data due to a misconfiguration. The network is eventually connected to a computer that provides access to the hosted service.The third layer is **Compute** which represents the platform, software, or operating system that hosts the business application; for example, Linux OS. In cloud technology, the compute layer could have a stack of technologies such as virtual machines or containers. Each computes virtualization technology should be assessed differently, For example, virtual machines run Linux OS as a hosting platform, then top up with a number of different virtual machines with different OS, whereas container-based systems run a single OS with different runtimes [25]. Each deployment should be secure against any attack, where any manipulation in hosting OS drivers could lead to disrupting all guest VM or containers. Once data are received by the compute, the next action is to hand over the data to the installed business service.The fourth layer is the **Application** that includes a single or a set of software. At this stage, data moves from transit to process or store. The application layer could be exposed to many threats due to the direct interface with business and external networks, unlike the compute or network layer where they are not meant to interact with business users.The final layer is **Storage** which describes the final destination for data, where the dataset is to be stored—from process to at-rest state. Storage can be local (attached disks) or remote (network storage). When data is stored in remote storage, threats to data while transitioning or in the process of network storage exist.

#### 4.2.2. Threat Actors

The actor is a term that refers to any human or machine attempting to access an organisation’s digital resources. The actor may be authorised or unauthorised, and the aim may be benign or malicious. Our approach assumed four types of actors:Business-User represents any human who is legitimately accessing the organisation’s resources and is looking for a specific service benefit.Business-Operator represents any human who has legitimate access to resources for administrative duties such as updating, maintaining, or troubleshooting.Business-Systems represent any service or process that is legitimately connected or able to connect to the organisation’s resources for business or administrative support, such as IoT.Threat-Actor represents any of the types (user, operator, or system) with the intention of abusing, disrupting business operations or gaining illegitimate access to data.

In this approach, the actor is a significant part of the threat analysis process, where the actor might be internal or external to the business. Depending on the actor and its features, it is necessary to determine potential attack scenarios that could disrupt the business. For instance, if a threat-actor gains the features of a business-operator with privileged access to organisational assets, the risk might be significant, including asset takeover and data infiltration, while if the threat-actor does not have privileged access, a real threat may not exist.

#### 4.2.3. Common Security Knowledge Base

Cloud threat analysis is a challenging task that requires various techniques for comprehensive threat analysis. d-TM advocates for the use of three widely used knowledge bases, including MITRE CWE which is useful for weaknesses, MITRE CAPEC for threats, and NIST SP 800-53 for threats analysis. Common Attack Pattern Enumeration and Classification (CAPEC) [26] entails organizing knowledge about adversary behaviour by focusing on a specific set of uses for system and application security. The knowledge base specifies typical approaches and properties that attackers use when exploiting security weaknesses in cyber-enabled capabilities such as clickjacking [27] and session fixation [28]. CAPEC is used to regulate attacker viewpoints and notions and standardise counter-security advances. The Common Weakness Enumeration (CWE) knowledgebase entails creating a collection of security software and hardware weaknesses that illustrate the necessary information to comprehend the nature of the system’s flaw. CWE is used by d-TM to govern recognised weaknesses and knowledge boundaries. Threats and weaknesses may be connected, classified, and compared for any given system using CAPEC and CWE IDs. Furthermore, information on the impact, mitigation, and associated dependencies is obtained. NIST SP800-53 publications are also adopted by d-TM, which provides a complete set of controls.

### 4.3. Threat Analysis Process

The process methodolgy used to analyse threats comprises of the following four main phases: data collection, data analysis, threat analysis, and threat mitigation. These phases consider d-TM concepts and it is worth mentioning that these phases are sequential and are necessary to identify, assess and manage the threats. Figure 2 represents an overview of the d-TM approach including requirements, stages and outcomes. The initial two phases provide the necessary knowledge of the business processes, services, and assets so that DFD can be generated to demonstrate the data flow in various phases. The threat analysis phase follows, in which assets and identified data are evaluated for weaknesses and related threats. Furthermore, discovered threats are prioritised in order to determine their threat criticality to business data. The last phase attempts to offer appropriate controls to reduce prioritised threats. The proposed controls are assessed for their security resilience in cloud-based computing organisations.

#### 4.3.1. Phase 1 Data Collection

This phase collects data relating to the organisational digital services, business logic and develops an inventory of assets related to the data that provides a thorough understanding of critical business services, and supporting infrastructure assets. It consists of two steps.


**Step 1—Understand Business Processes and Services**


The goal of this step is to define the business logic in terms of important business processes, digital services, and appropriate infrastructure. Each functional process in any organisation is backed by one or more digital services, as well as infrastructure that facilitates digital services.

Each selected service’s criticality must be assessed during the security analysis. Although the influence of each service on corporate objectives differs depending on specific organisational context. However, the characteristics and criticality of each service are determined through several options, such as stakeholder brainstorming or interviews. Each process (Bp) is specified using many attributes such as process information, related services (Bs), and its importance to business objectives. However, determined processes and services are considered within scope of work of the threat analysis process, where identified services are categorised as follows:High (H): mission-critical services that offer core functions to the business; consequently, the firm cannot function without it.Medium (M): supporting services to the core function, where business interruption is limited and can last for a short length of time.Low (L): a service that provides support to fundamental functions and allows the organisation to operate with minimum effect for an extended period of time.Non-critical (NC): a service that supports the business but has no influence on core functions, such as a customer satisfaction system


**Step 2—Understand Business infrastructure Assets**


This step identifies the infrastructure assets that support each critical service. The assets are classified according to their type, management techniques, and underlying dependencies. Interviewing technical stakeholders and acquiring design materials might be used in this approach. The information acquired might include system architecture, configuration files, logs, and other necessary details. As a result of this step, security analysts must identify which infrastructure supports prioritised business functions. For asset valuation, three properties are considered.

Asset Type: Information needed to characterise the asset, such as its name, model, and so on. In addition, each asset should be mapped to one of the five d-TM layers which are the agent (Agt), network (Net), compute (Cmp), application (App), and storage (Stg).Asset Administration: The access mechanism used by admins to operate the asset; security analyst needs to determine the protocols, tools, access conditions and required privilege to operate the asset.Asset Dependency: Inherent dependencies among the asset for service delivery. This property helps understand potential attacks that could be conducted because of a weakness in a dependent system which might result in a sequential failure or breach cascading impact. d-TM considers MITIGATE Classification [29] for dependency types and access mechanisms of assets; hence, dependency can be determined using the following two factors: Dependency Type and Access, where dependency type defines the relationship between two assets. There are five types of dependency among assets (host, exchange, storage, control, or process). The Access mechanism represents how an asset communicates with another asset, there are three types of access direct, local and remote.

#### 4.3.2. Phase 2 Data Analysis

In the previous phase, critical business services and supporting infrastructure are identified, and business logic is realised. The steps within this phase are discuss below:


**Step 1—Identify and Extract Data-levels**


This step focuses on the information gathered in the previous phase, which is about how data is operated per asset to the related services. Security analysts must have the necessary expertise to comprehend technological configuration, codes, and designs that interact with data. When extracting information, the d-TM notions such as the actor, layer, and data level should be considered. This information is recorded into a table as a single portion of configuration or code lines that describe a specific function; additionally, descriptive language may be an option for systems that provide a configuration in a graphical interface, such as cloud portals, windows-based cloud computing, and some applications. The descriptive alternative could be straightforward and fair in providing insights to organisation technicians, developers, and non-expert matters managers about how assets are configured to run data. The exercise should be security-driven and associated with data processing, transmission, or storage. For example, a cloud compute running a Linux operating system that may be accessed using the ssh protocol for management needs. In this case, it is clear that the data are management-level, the actor is the business-operator, and the necessary information or configuration to extract is related to the ssh authentication technique, authorization level, protocol version, and access port among other characteristics. Furthermore, modern cloud apps increasingly depend on API calls. Application API configuration might provide suitable data with which to extract control data where business-systems communicate for data or attribute exchange. Table 1 shows the organisation’s security team where business data are operating and at what phase it may be at any point of the infrastructure.


**Step 2—Construct Data-flow Diagram**


DFDs are one of the most commonly employed techniques in threat modelling because they are simple and do not really require any technical understanding of software design [30]. This step intends to use the results of the previous steps to create a data flow diagram for the prioritised services. The model makes use of the d-TM principles such as data levels at each asset, actors, and the layered technique to utilise the common symbolic representation of DFD. This modification increases the utility of the diagram for comprehending data rather than just flow. As a result, the data are visualised into three levels and phases.

#### 4.3.3. Phase 3 Threat Analysis

This phase outlines the threat assessment activities in two steps. The output from this phase is documented as the threat profile knowledgebase.


**Step 1—Identify Weaknesses and Associated Threats**


This initial step investigates the collected data at each data level to identify weaknesses that could lead to potential threats. Security analysts could use a manual code review and architecture or design review as a detection approach to identify weaknesses in the acquired information. However, in most circumstances, the second technique provides highly cost-effective and comprehensive coverage for networked assets. Nonetheless, expert judgment is necessary to identify suitable weakness identifiers because it is impossible to be familiar with all technologies. Overall, the examination seeks potential identifiers by comparing the acquired data to potential risks when assets store, process, or transfer data at each data level. d-TM considers the CWE knowledgebase by MITRE [31] as a reference of weaknesses. CWE provides multiple views that could help expert matters to find matching weaknesses; the views include software, hardware, research concepts, and other criteria. Once weaknesses are identified, findings are recorded in a table according to CWE IDs for each data level and phase.


**Step 2—Prioritise Threats**


This step aims to determine the criticality of each threat to the business so that appropriate actions can be taken to mitigate the threats. Threat priorities are determined using a novel approach that considers multi-factor evaluation, including Business-as-Target (Bt), Threat-Complexity (Tc), and Business-Impact (Bi). The overall priority is determined based on the correlation of the three factors of five threat scales: very high, high, medium, low, and very low.

Very High: A very high impact on business continuity and critical business services which requires immediate action.High: A significantly high threat to business continuity, and critical business services are interrupted which requires attention in a specific time frame.Medium: An intermediate threat to business continuity, and no critical business services are interrupted due to its impact. Furthermore, some supporting business services are impacted, and business can run for longer. Additionally, the apply action should be completed within a timeframe of a year.Low: A low threat to business continuity, and no critical business services are interrupted due to its impact with option control. Furthermore, some supporting business services are impacted, and the business can run normally.Very Low: A significantly low threat to business continuity, and no critical business services and supporting services are interrupted due to its impact. Additionally, applying action could be optional or ignored.


**Factor 1 Business-as-Target (Bt)**


The first factor refers to the likelihood of an organisation being highly targeted by an attack, which can be identified based on a correlation of the threat-occurrence metric and the attacker-gain metric. Although threat-occurrence represents the likelihood of a specific threat occurring, the value considers the organisation’s attack history, as well as public threat awareness records, e.g., the IBM X-Force Threat Intelligence Index [32]. Likewise, attacker motivation could be determined using a public threat record based on business context and the attacker’s performance in that business sector. Two metrics for Bt and the matrix in Table 2 are as follows:Metric 1: Threat-Occurrence represents the probability of a particular threat occurring for an asset with three scales:○High (H): The organisation experiences this threat twice or more in a one-year timeframe. Industry researchers forecast this threat as a top-rated attack for similar businesses within one year;○Medium (M): The organisation experiences this threat once in a one-year timeframe. Industry researchers forecast this threat as medium-rated attack for similar businesses within one year;○Low (L): The organisation experiences this threat once or not at all in a two-year timeframe. Industry researchers forecast this threat as a medium to low-rated attack on similar businesses within the last two years.Metric 2: Attacker-Gain represents the goal behind the attack including curiosity, personal gain, personal fame or national interests with three scales including high, medium and log.


**Factor 2 Threat-Complexity (Tc)**


At this step, the complexity of a particular threat is determined as high, medium or low. The correlation matrix is based on Attacker-Capability and Access-Complexity. The two metrics and correlation matrix are defined below:Metric 1: Attacker-Capability refers to overall attacker capabilities such as skills, knowledge, opportunities, and resources that the attacker incorporates to exploit a weakness. Capabilities could be estimated as high, medium or low.○High: A sophisticated level of expertise and knowledge with adequate resources to generate opportunities for continuous attacks.○Medium: Moderate level of expertise and knowledge with reasonable resources to provide a considerable ability to generate multiple opportunities and continuous attacks.○Low: A low level of expertise and knowledge with limited resources and the ability for attack.Metric 2: Access-Complexity determines the level of complexity needed to exploit a particular weakness, where each organisation has different levels of access and controls. Attackers often estimate the level of complexity of any attack to find the easiest and success-guaranteed approach to compromise organisation data. Likewise, security analysts evaluate existing access mechanisms and implement security controls for identified gaps; it is important to understand the access complexity that could reduce the likelihood of exploiting existing weaknesses. Access complexity can be estimated as high, medium or low. Table 3 represents the correlation matrix for the likelihood of threat complexity for a particular asset.○Multi-level Access: The attacker requires a special access condition, and this condition requires a high level of effort and expertise that could include a multi-stage attack.○Single-level Access: The attacker requires a somewhat special access condition, and this condition requires a medium level of effort and expertise.○Direct: The attacker requires no special access condition.


**Factor 3 Business-Impact (Bi)**


The final factor determines the business impact and threats priority is calculated by correlating the outcome of Factor 1 (Bt) and Factor 2 (Tc) with business impact (Bi), whereas the impact probability to a business is an essential factor, where impact to the business is classified as high, medium or low. However, the result of overall correlation is shown in Table 4.
○High (H): The expected impact of the mission-critical services that provide core functions to business is High; the business cannot run without it.○Medium (M): The expected impact to the business is medium, where the supporting service is impacted; the business could run for some time.○Low (L): The expected impact to the business is low, where businesses run with minimal impact.


#### 4.3.4. Phase 4 Threat Mitigation

This final phase aims to determine the suitable controls to mitigate identified threats and ensure overall security assurance.


**Step 1—Determine Controls**


This step intends to provide appropriate controls based on the identified threats per data. The controls are based on the vulnerabilities that could be exploited by a threat. While threats are materialised due to a weakness in the system, it is essentially advocated to evaluate suggested mitigation guidelines by CWE to identify the controls. In order to understand the details behind identified weaknesses, security analysts could use CWE guidelines to provide the required information about each weakness, part of this information is “Scope” and “Impact” properties which are available under the “Common consequences” section of any weakness. Additionally, the security analyst should find the required mitigation technique to overcome this weakness. To determine the direction of potential mitigation techniques, another section from CWE was investigated, the section called “Potential Mitigations” which is presented in the textual format of techniques required to mitigate identified weaknesses. Based on the essential scope understanding of the potential controls that are determined, security analysts must enumerate each relevant NIST security control family with the support of scope, impact, and mitigation techniques provided by CWE guidelines.


**Step 2—Determine Assurance-level**


This step determines the assurance level of overall cyber security taking into account the identified threats, controls and data. The assurance level is based on the completeness, effectiveness and complexity of the security control to tackle the threats to overall data security. The three factors are correlated to determine the overall assurance level of the control. The overall assurance level (OAL) is determined based on Equation (1). Where the Assurance level is High when OAL value = (7–9), Moderate (4–6), Low (less than 4).
OAL = Ct + Ef + Cx(1)

Completeness (Ct): The identified controls are relevant to tackling the threat and related weaknesses. Controls are evaluated specifically in terms of the level of coverage that results in the elimination or reduction of threat impact. Controls can provide none, partial, or full coverage to a specific threat, whereas control capabilities are designed to mitigate threats to specific data levels at specific phases, such as attacks to management data levels in the transit phase.○High (H = 3): Control provides the necessary features to mitigate the threat’s likelihood without requiring any additional enhancements or supporting controls.○Medium (M = 2): Control provides some features to reduce the threat likelihood; however, additional enhancements or supporting controls are required.○Low (L = 1): Control provides a significantly minimal feature to reduce the likelihood of the threat; however, additional enhancements or supporting controls are required.Effectiveness (Ef): Effectiveness is the ability of security controls to effectively protect, detect, and respond to threats. This implies it must be capable of preventing the threat from occurring in the first place, recognizing it when it does, and successfully responding to limit the impact.○High (H = 3): Control aims to prevent the occurrence of an attack, as well as to detect and respond, when necessary, without the need for any further enhancements or supporting controls.○Medium (M = 2): Control aims to provide two essential roles, such as protecting and detecting the occurrence of attack, with no response. However, further enhancements or supporting controls are required.○Low (L = 1): Control aims to provide a single essential role such as detecting the occurrence of attack, with no protection and response. However, further enhancements or supporting controls are required.Complexity (Cx): When it is difficult to implement or operate control, a number of challenges to the organisation’s security team become evident. To avoid major changes to the overall system, control should be smoothly integrated into organisational systems. Furthermore, operation complexity presents a challenge when the team is unable to operate the control due to a lack of knowledge or because the control provides a complex workflow to apply actions.○High (H = 1): Control can integrate seamlessly into organisation infrastructure and the team have the skills to implement and operate the control.○Medium (M = 2): Control can be integrated into organisation infrastructure with minimal changes, and the team have no skills to implement and operate the control.○Low (L = 3): Integrating control in organisation infrastructure is a complex procedure and the team lack the skills to implement and operate the control.

## 5. Evaluation

To demonstrate the applicability of the d-TM, we employed the proposed d-TM in a real-world use case scenario. The scenario concerns a cloud-enabled corporation looking to grow its business revenue and improve its operational experience by leveraging Google cloud services. This section presents the implementation of d-TM and observations about using d-TM to analyse the threats. The purpose of the evaluation is as follows:To understand the applicability of using d-TM;To identify any issues relating to the implementation of d-TM.

### 5.1. CaseStudy Scenario

The scenario is about a fast-food chain restaurant located in the Middle East. It was founded in the early 1980s and now employs 5000 people in over 300 locations. Due to confidentiality reasons, we cannot disclose any identifiable information about the organisation. On the SAP S/4HANA [33] (on-prem) platform, the firm manages restaurants, trucks, and material orders, as well as the supply chain. To protect its infrastructure, improve performance, decrease downtime, and develop a disaster recovery strategy, the company moves its infrastructure that supports sales operations (SAP platform) to the cloud. As the organisation is looking to establish more systems and freedom of control over their assets, they find that the IaaS cloud model is a suitable cloud option. With S/4HANA on Google Cloud, system stability is improved, monthly financial reports are extracted in half the time, and IT helpdesk enquiries are cut in half. In order to keep corporate infrastructure and data in the cloud safe, the organisation tries to identify threats that could occur as a result of cloud migration and solutions that could help. Figure 3 presents the overall architecture of the organisation which includes internal and outsourced infrastructure. Sales operations are hosted by Google Cloud and accessed by the sales team from local networks and restaurants. Each restaurant is connected to a wireless network that is connected to an internet modem. On other hand, finance and management access the cloud using corporate internet; likewise, corporate cloud admins can also access the cloud using corporate internet. The cloud platform consists of a virtual firewall that acts as a cloud gateway for sales operation applications. The application in the cloud runs using three cloud computing instances, namely the SAP application NetWeaver, DB and storage. However, the remaining services, on the other hand, are located in the local data centre. The focus of this evaluation was on the sales operation outsourced to the cloud, which is the company’s most important service.

### 5.2. d-TM Implementation

The outcomes of applying the d-TM model to the case study are presented below. However, d-TM does not deploy any infrastructure, instead it only analyses threats, weaknesses and related controls. This section represents the flow of the process as well as the information generated by each phase. A team was formed with the first author and two employers of the organisation for the purpose of implementation.

#### 5.2.1. Phase 1 Data Collection

Based on the understanding of the existing business logic knowledge and interviews with the key personnel including the head of infrastructure, network and system admins, and branch manager who uses the PoS machine at one of the branches, the first step aimed to identify the critical business services and underlying infrastructure. Sales operation is a core function of the company, and is built based on the SAP S/4HANA platform hosted by GCP. The platform is also accessed and operated by the company IT support team. The upper part of Table 5 shows details of the sales operation and the lower part represents the underlying assets supporting the SAP platform and technical specifications. The table also provides a sample list of assets detected by d-TM.

#### 5.2.2. Phase 2 Data Analysis

Once the information relating to the business process is gathered, then it is necessary to identify the related data that support the identified business process. This phase aims to discover useful information that indicates how an asset handles data at every level and phase in addition to data presentation while moving across platforms. The identified actors are presented below:Business User (aka USR), which refers to any staff of sales, finance, or restaurant front-end representatives that use Point-of-Sales machines. This actor has limited privilege and is able to access the SAP application for business-related activities such as registering or monitoring sales orders.Business Operator (aka OPR), which refers to any IT Staff working for the organisation that access infrastructure or services for any administration activities such as troubleshooting. This actor has a high privilege that provides full control of the asset.Business System (aka SYS) which refers to any system-to-system relationships, such as the SAP application and SAP DB as well as CSP automation controllers and organisation compute. It is a system or process of exchanging data using API calls for instance. The system may have full or restricted privileges, depending on its role. For example, the CSP console has full privilege over organisation compute hosted in the cloud.

Collected data at the previous stage are analysed by the team, and the outcome is driven by the understanding of each asset, actor type, data level and phase. Table 6 depicts the Bs0. The first row represents the analysis outcome, while business users (USR) generate business data (bD) by accessing the service that is hosted on the cloud using chrome browser (Agt0) installed on the PoS machine; data are shown at each phase as well as the way they are stored (Dr), processed (Dp) and transmitted (Dt). As result, the browser runs default configuration and open-source plugins that are not used for business purposes, which shows no governance over the existing configuration to control software features such as storing sensitive data about session information locally with no protection, or the presence of malicious code that interferes with the browser function such as intercepting business data or escalating the attack to gain access to the local system. Eventually, this could lead to a threat to data while at rest (Dr) or processing (Dp). On the other hand, business data are sent over a wireless network that is also shared by the branch staff, who use their personal devices to access the internet. The data while transmitting (Dt) is at risk due the potential of compromised devices being connected to the same network.

Figure 4, represents the business service (Bs0) data flow diagram, that shows a business user using a PoS machine to access the sales operation application that is hosted in the cloud. The figure shows that when the sales person uses web browser at PoS machine to access cloud hosted services, the data traverse over multiple assets to the final destination. The figure shows that the data levels and phases at each asset facilitate sales operation application.

#### 5.2.3. Phase 3 Threat Analysis

This phase focuses on the threat analysis for the identified critical business service (Bs0). That includes identifying weaknesses, related data threats and the criticality of identified threats to data. Organisations turn to the cloud for agility and scale-on-demand as they modernise or convert their SAP systems (on-prem) to SAP S/4HANA (Cloud). Adopting additional cloud services or managing hybrid environments expands an organisation’s threat surface. Security threats are aimed at the SAP Application, the online interface, and PoS devices that link to SAP. Cyber threats employ infrastructure as a point of entry to gain access to SAP’s sensitive data. SAP does not typically provide infrastructure security guidelines, and SAP’s Security Baseline Template leaves these issues to the client to resolve [34]. As a result, d-TM considers this gap and provides the necessary threat assessment to the SAP platform and its surrounding infrastructure.

The threat analysis process begins by identifying the weaknesses of the data by following the CWE reference model. A total of seven weaknesses were identified which were relevant to the scenario as shown in Table 7. The evaluation considered two infrastructure assets (Net0 and Net3), SAP front-end application, and two different agent softwares (Agt0 and Agt1) that were used by the organisation’s users to access or operate cloud infrastructure. The d-TM approach begins by assessing infrastructure from the user end, which is subject to many threats. For instance, the business user uses a web browser (Google Chrome) that has an asset id (Agt0) to access the SAP application, during which the Agt0 is assessed to identify weaknesses by evaluating the existing configuration, user practices and hosting operating system. This results in the following:The browser runs with the default configuration.Some commercial plugins are installed that are not related to business.A single profile is used by multiple users to access the windows operation system.The PoS machine uses an insecure wireless network, the network is accessed and shared with restaurant staff for personal use.The PoS machine data stored in the windows operating system are accessible by restaurant staff.

The first row of Table 7 indicates the weaknesses discovered during the inspection of (Agt0). The weaknesses are primarily demonstrated as an uncontrolled asset, which provides a risk to all data phases (Dr, Dp, Dt) for the business data level (bD), as well as to management data (mD), if an administrator uses this system to access sensitive data or do maintenance. The first weakness demonstrates that the browser and the organisation lack a security control or policy to prevent or evaluate any program installed on the browser. Additionally, the information from browser sessions, history, and saved passwords are stored locally with no access restrictions or encryption, thus putting stored data (Dr) at risk. Following the CAPEC reference model, the model determines corresponding threats to the data once the weaknesses are detected. As shown in Table 7, there are a total of seven threats that are relevant to the scenario. In addition to threat identification, the process aims to determine the criticality of each identified threat. The criticality of each threat is determined by the d-TM process using the correlation of three factors (Bt, Tc, and Bi). For example, based on the organisation’s security engineer, the adversary in the browser attack shown in Table 7 does not often occur but it was experienced by the organisation previously at multiple restaurants. This attack could provide an attacker with a moderate gain by inspecting the traffic that is generated by the PoS machine regarding the sales activity of the branch, and could also make the salesman credentials vulnerable to risk, which is considered as having a low impact on businesses due to the restricted privilege assigned to PoS sales representatives’ accounts. However, this weakness could provide high gain to attackers and have a high impact on the business if the browser is used by organisation system admins. In terms of the information accessed and considering that the browser is used by salesmen and not the admin, the overall criticality of this attack is Low (L), where Bt (M), Tc (M), and Bi (L). Lastly, the threat analysis of (Bs0) data is presented as a threat profile table which contains information on weaknesses, threats and their overall priority as determined by the d-TM threat analysis technique. Table 7 depicts the format of the threats profile table, which is formed of various columns. The columns aim to represent a list of assets evaluated and mapped to their weaknesses and the introduced threats. Threats and weaknesses are listed with descriptions and identifiers for CWE and CAPEC catalogues. The reason for using catalogue identifiers rather than custom-generated ones is to provide an industry understandable language and reference. Additionally, the table provides the organisation with the criticality of each threat so that a mitigation strategy can be devised.

#### 5.2.4. Phase 4 Threat Mitigation

While threat is determined by the system’s weaknesses, d-TM simply suggests an evaluation of CWE’s suggested mitigation procedures to establish controls. d-TM perceives detected weaknesses as a source of threats that must be addressed. The “Scope” and “Impact” features, which are provided under the “Common consequences” section of every weakness identifier, could provide needed information about any weakness. Moreover, CWE “Potential mitigation” aids in the comprehension of the required mitigation plan. These details provide d-TM with a better idea of what the attackers stand to gain with this attack and what kind of control they require. At this stage, however, specialised expertise is required to correlate these data to appropriate NIST controls. As a result, security analysts must enumerate each relevant NIST security control family using CWE standards’ scope, impact, and mitigation techniques. Table 8 represent the required mitigation controls based on the Bs0 threat profile. The table includes threats that required immediate attention according to their severity. However, the threats assigned a Very-high (VH) score were addressed at this phase. The table focuses on four threats that required mitigation which target cloud infrastructure elements (Net3 and App0). The threats can be assessed as being directed towards organisational data by account takeover, D/DoS, and application input misuse. As a result of understanding the implied weaknesses, controls should be implemented within the scope of identity management, resource management, and system data integrity by mapping these scopes to the NIST controls family, the NIST IA (Identification and Authentication), SC (System and Communication Protection), and SI (System and Information Integrity) controls family.

Lastly, the table summarizes the assurance level analysis of each control in relation to risks for a specific type of data. For example, the control (IA-2) is designed to protect management data (admin credentials) from a Credential spoofing attack caused by a lack of strong authentication requirements. A multi-factor authentication (MFA) solution was recommended by the control. The examination of this control revealed that the solution provides a significant mitigation match to a specific threat, which MFA considers to be exceptionally successful. MFA, on the other hand, is sometimes difficult to implement due to the necessity for integration with vital systems. The outcome of evaluating this control is an 8 rating, indicating that control assurance for the firm is very high.

## 6. Discussion

The application of d-TM in the studied context is promising. This section provides our observations after implementing d-TM in the studied context.

### 6.1. d-TM Process

The process begins with an understanding of the business logic, which highlights the criticalities of existing systems to business continuity and narrows the efforts to low-priority business services. The analysis reveals that the main process for business revenue is sales operation, which is implemented on a cloud-enabled service platform. This important service is the SAP solution. In terms of business logic comprehension, the study used DFD diagrams to demonstrate the relationship between corporate cloud-enabled systems and company endpoint assets. The flow diagram was provided to ease in the comprehension of how the underlying technology interacts with data directly or indirectly. The actor scenario described in DFD indicates that business users utilise PoS computers running a web browser as an agent to access the cloud service application (SAP), where the request is initiated by the agent, the data traverses over numerous network elements, locally and remotely (cloud), before being handed over to the business application hosted on cloud computing and leveraging cloud storage. DFD components for various actors were evaluated in order to extract and analyse data based on the d-TM concept. The goal of this assessment was to evaluate acquired data about running infrastructure and identify relevant information about how each asset operates with the data. The asset evaluation of the studied context was carried out based on the created DFD, which presents information that could lead to a weakness in the asset, resulting in a data threat. For example, Agt0 and Net0 are two critical components for running business applications in any branch; they run with basic configuration and are not governed by the company. Furthermore, App0’s authentication methodology for business users relies on user names and passwords. Similarly, Net3 is required for operator access.

### 6.2. Threats and Control

As previously indicated, the company is moving to the cloud to upgrade its on-premises systems to the cloud. Cyber-attackers use infrastructure as an entry point to target sensitive data. In contrast, the SAP application does not provide infrastructure security recommendations or a security baseline template, leaving these concerns for the customer to resolve [34]. The threat analysis approach was designed to examine information and DFD diagrams derived from the data collection phase. The study provided a sample of detected weaknesses, which were as follows:**Assets**: A total of five assets were investigated and regarded as significant components in the cloud data path. Data were created either by the business user at the PoS machine or by the system administrator using a corporate laptop to access the cloud. The assets are Ag0, Ag1, Net0, Net3 and App0.**Weaknesses**: A total of seven weaknesses were determined out of the five assets. The seven weaknesses were investigated to reveal the implied threats to data.

Overall, the weaknesses revealed that the organisation does not sufficiently consider the restaurants’ endpoints and network when it comes to security. As a result, endpoints run an unregulated Chrome browser and operating system, thereby placing company data at risk whether it is created or saved locally or in the cloud by salespeople. System administrators, on the other hand, share the same weaknesses as users who use uncontrolled browsers and support IT tools such as “Putty.” This could be used to store sensitive information regarding an organisation’s operational data, such as the address of its servers on system administrators’ laptops. These weaknesses could have an impact on company data when in transit, in process, or at rest. An attacker might use the flaws to intercept, manipulate, or exfiltrate data in the browser. Furthermore, an attacker may employ a malicious plugin to obtain access to system files or conduct a browser MITM attack.

With regard to data threats, the assessment discovered multiple weaknesses that could be utilised by an attacker to impose threats to data while in the cloud. As a result of the d-TM approach, multiple threats were discovered in reference to the identified weaknesses.

**Threats:** A total of seven threats were identified out of the five assets assessed. The threats targeted data at infrastructure components in different layer such as agent, network, and application.**Critical threats:** A total of four threats were categorised as critical out of seven identified threats. The four threats required immediate attention in the case study. Denial of service, XSS, and social engineering represented critical threats due to weaknesses in two assets (Net3, App0).**Data:** A total of two cloud assets out of five represent a high risk to data; where threats mainly target two data levels which are management and business data levels. Furthermore, data are at risk while in two phases which are transit and in process.

The research revealed multiple possible threats within the cloud infrastructure, such as virtual cloud instances including application front–end interface, namely Net3. This device was configured to accept any request and provided no means of controlling the volume of incoming traffic. This could be due to a constraint in the asset’s capabilities or the assumption of the presence of a third-party security solution. In our scenario, no solution was available. As a result, D/DoS attacks were made possible on these internet-facing assets. Cloud business services could be severely disrupted as a result of such an attack. This issue was rated as a very high threat by the d-TM, and must be addressed as quickly as possible. Furthermore, the front–end cloud application lacks validation capability for incoming requests, making the application vulnerable to a Cross-Site Scripting (XSS) attack. This attack was also a high-ranking one. Finally, cloud apps and network assets rely on credentials that users memorize; this weakness could lead to a variety of threats aimed at users, such as social engineering attacks. On other hand, if a business user’s credentials are taken, the impact is limited; however, this is not the case if the credential belongs to a system administrator. Therefore, when the system administrator is an actor, the threat to management data is rated as very high. 

The d-TM model provides the organisation with appropriate controls to help mitigate the identified critical threats.

Controls: Four controls were identified to mitigate the critical threats. The controls include looking after authentication improvement, input validation and service assurance.

The organisation mostly faced attack from threats that target system identification mechanisms, resource capacity management, and application input control, according to the high critical threats presented in the threat profile. As a result, the model suggested a set of measures to help organisations deal with the threats, such as implementing multifactor authentication mechanisms, implementing a validation control on user inputs, and obtaining technology to monitor and respond to resource misuse.

### 6.3. Comparing with the Existing Works

We compared the findings from the studied context with the existing works in the state of the art. Threats to an organisation’s cloud infrastructure may be comparable, particularly if they use the same cloud paradigm, such as IaaS. The OWASP “Top 10 Cloud Risks” [39], highlighted two issues revealed by the d-TM analysis method, “User Identity Federation” and “Infrastructure Security”. Furthermore, Denial-of-Service (DoS) attack and threats relating to identity management are also considered in [2,40,41]. The consequences of a cloud account takeover are even more difficult to trace and mitigate. However, the majority of the similarities in detected threats were related to cloud applications or systems. Moreover, the analysis from the studied context helped to uncover additional threats that were not covered by others which are related to underlying infrastructure such as Agt0, Net0 and Net3 in our scenario. d-TM includes the DFD approach with seven weaknesses based on CWE, seven threats described by CAPEC and three NIST controls. All were identified from the studied context.

In summary, the d-TM presents significant value for the current threat modelling practice due to many features that empower the model. The model considers business processes and services as the initial reference points with which to analyse threats, which is overlooked by existing methods. It also defined attacks and data levels for the threat analysis. d-TM provides additional value compared to others models that maintain attack surfaces for expert judgments, which could result in inconsistency in reproducing results or lead to overlooking a necessary attack surface. As a result of business processes, services and layered infrastructure understanding, d-TM translates information to a data-driven DFD diagram that presents data types and phases of any infrastructure asset. A data-driven leveraged DFD provides an important advantage to any organisation, as traditional applications or asset-based DFDs used in most research lack the feature of data level and phase. The threat analysis process considers weaknesses identification based on CWE KB, which was later used to determine implied threats to data, and threats also take advantage of CAPEC KB. d-TM provides a new way to identify the criticality of each threat; the criticality is determined using three necessary factors, namely Bt, Tc and Bi. These factors enable d-TM to integrate business understanding into the prioritization process. Lastly, the identified critical threats were evaluated to determine suitable controls and it is assurance to business objectives. None of the discussed research considered three data levels, data-oriented DFD, three actors use cases, five attack layers (particularly Agent) and common KB to analyse data threats. However, d-TM is based on data so the identified risks were both generic and unique in our case. Table 9 shows a qualitative comparison of d-TM with the existing threat models.

### 6.4. Limitations of d-TM

We identified a number of limitations of d-TM. Firstly, threats were specific to one cloud service model/multiple identified services. It is difficult to draw DFD for a complex business process that links with multiple layers and CSP. In the application of our work to the case study, it was difficult to understand whether the CSP deployed the recommended controls to safeguard user data in the CSP infrastructure. Furthermore, SaaS and Function as a Service (FaaS) (i.e., serverless) cloud models are difficult to follow due to the DFD’s layered approach, as these cloud models are constructed on a single layer. Because the underlying infrastructure is buried under the authority of the CSP, companies are not able to check data levels and phases at any point in the cloud.

## 7. Conclusions

This paper presented a novel threat analysis approach supporting the identification, analysis and control of threats from three data abstraction levels and phases to secure the overall cloud-based ecosystem. The d-TM considers five different layers in cloud-based systems to support the threat analysis. The process provides an understanding of organisational business processes and services so that data within these processes can be identified and analysed based on possible weaknesses. Hence, an understanding of business infrastructure assets helps not only to develop data flow diagrams for the infrastructure between user and cloud service provider but can also help to identify possible threats that could pose any risks to the system context. d-TM was applied to a real cloud-based use case scenario, and the results show that threats from the system can be identified using our approach. Future work will focus on improving the d-TM process to address the identified limitations. Moreover, our efforts will be directed towards deploying the d-TM to different use-case scenarios to generalise our findings. We also plan to automate the process by developing a d-TM threat analysis platform.

## Figures and Tables

**Figure 1 sensors-22-05726-f001:**
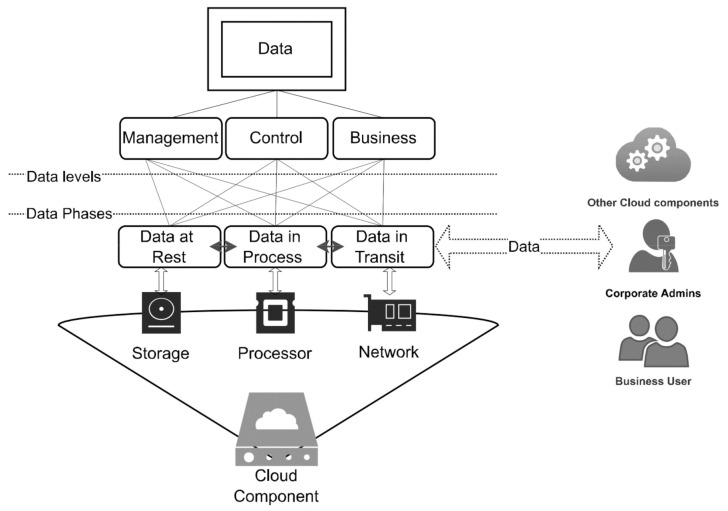
Data levels and phases.

**Figure 2 sensors-22-05726-f002:**
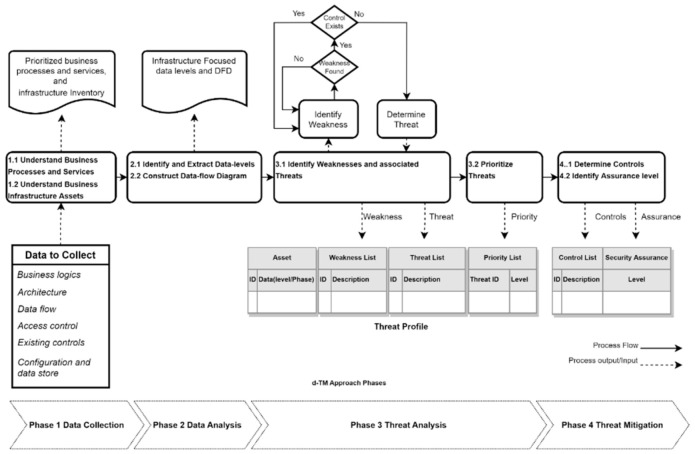
An overview diagram of the Data-driven threats analysis approach.

**Figure 3 sensors-22-05726-f003:**
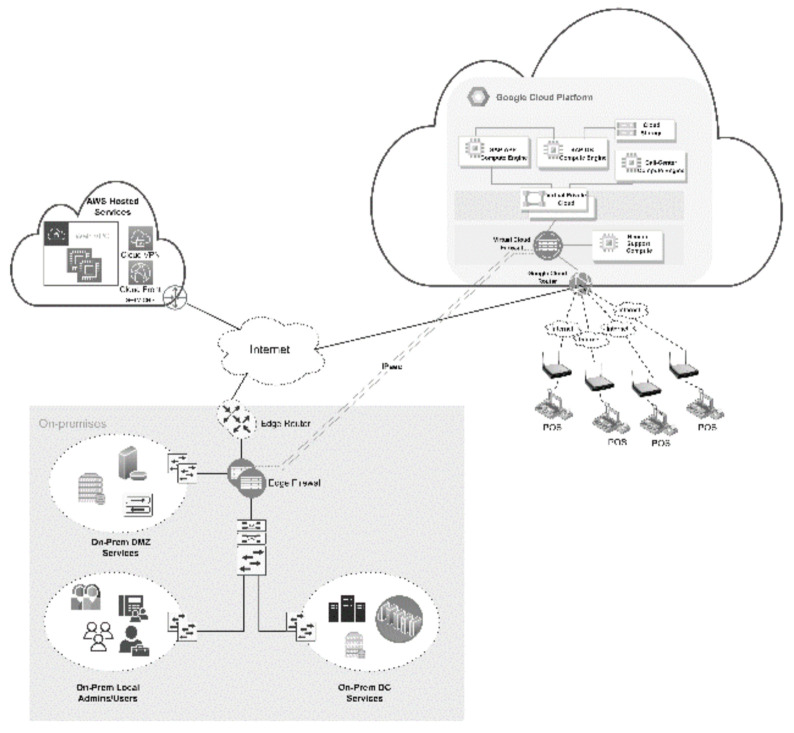
The scenario technology architecture.

**Figure 4 sensors-22-05726-f004:**
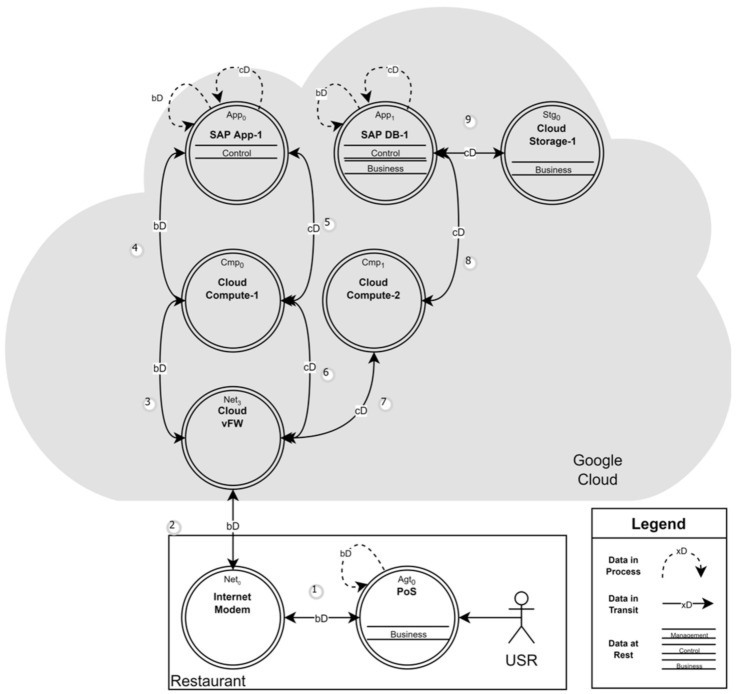
A data flow diagram of a business user accessing the cloud application.

**Table 1 sensors-22-05726-t001:** The table represents the data level and phase at any asset.

Business Service (Bs)	Actor	d-TM Layers	mD	cD	bD	Dr	Dp	Dt
**Bsx**	Business-user (USR)	Agtx			✓	●	●	●
		Netx			✓			●
		Cmpx			✓			●
		Appx			✓	○ ^1^	●	●
		Stgx			✓	○ ^1^	○ ^1^	○ ^1^
**Bsx**	Business-Operator (OPR)	Agtx	✓			●	●	●
		Netx	✓			○ ^1^	○ ^2^	○ ^2^
		Cmpx	✓			○ ^1^	○ ^2^	○ ^2^
		Appx	✓			○ ^1^	○ ^2^	○ ^2^
		Stgx	✓			○ ^1^	○ ^1,2^	○ ^1,2^
**Bsx**	Business-System (SYS)	Agtx		✓		●	●	●
		Netx		✓				●
		Cmpx		✓				●
		Appx		✓		○ ^1^	●	●
		Stgx		✓		○ ^1^	○ ^1^	○ ^1^

*Where*, “x” refers to service or asset id. “✓” refers to presence of the data level. “●” refers to the presence of the data phase. “○” refers to a potential presence of data, the presence depends on other factors. “^1^” The Data is stored local (locally attached disk) or remotely, while remotely it needs data to be sent, processed and stored at network storage. “^2^” The asset which the operator intends to access.

**Table 2 sensors-22-05726-t002:** Business-as-target matrix.

Bt	Threat-Occurrence Likelihood		
Attacker-Gain Scale	High	Medium	Low
High	H	H	M
Medium	H	M	L
Low	M	L	L

**Table 3 sensors-22-05726-t003:** Threat complexity matrix.

Tc	Access-Complexity Levels		
Attacker-Capability Levels	Multi-Level	Single-Level	Direct
High	M	L	L
Medium	H	M	L
Low	H	H	M

**Table 4 sensors-22-05726-t004:** Table represents the Matrix of threat priority.

Threats Priority		Bi								
		High			Medium			Low		
	Tc	Low	Medium	High	Low	Medium	High	Low	Medium	High
Bt										
High		VH	VH	H	H	H	M	M	M	L
Medium		VH	H	M	H	M	L	M	L	VL
Low		H	M	M	M	L	L	L	VL	VL

**Table 5 sensors-22-05726-t005:** Table of critical business processes, services and assets.

Process ID	Process Name and Description	Relevant Service ID	Service Name and Description	Service Criticality
Bp0	**Sales Operation** This process is critical to the company’s business continuity, where sales operations are the main source of revenue for the company.	Bs0	**SAP Platform** This system provides a platform for sales representatives performing day-to-day sales tasks such as sales, checkout, balance, and purchasing. On the other hand, it provides management with information about all sales data and needs for planning and supporting the sales process.	H
**Bs_0_**				
**Asset details** *Asset_id_ (Name/Role*, *Brand name*, *SW version)*		**Asset Administration** *Mgmt. (port, agent*, *access*, *privilege)*	**Asset Dependency** *Dep. (asset*, *type*, *access)*	
Agt0 (Web Browser, Google Chrome, 101.0.4951.54)		-	-	
Agt1 (SSH Terminal, Putty, 0.74)		-	-	
Net0 (PoS Internet Modem, ZNID, S3.1.135)		Mgmt. (443, Agt0, W/LAN, Local Admin)	Dep. (Internet link, Exchange, Direct)	
Net3 (Cloud vFW, Fortigate-VM, 5.4)		Mgmt. ((443, GCP-Console, GCP-Shell), Agt0, LAN, Security Admin group)	Dep. (GCP GW, Exchange, Local Network)	
			Dep. (Jump-Server, Exchange, Local Network)	
			Dep. (SAP Application, Exchange, Local Network)	
			Dep. (GCP Cloud Portal, Control, Local Network)	
Cmp0 (Cloud Compute-1, SUSE Linux, 15)		Mgmt. ((22, GCP-Console, GCP-Shell), (Agt1, Agt0), Remote (Internet), GCP Admin group)	Dep. (SAP Application, Process, Direct)	
			Dep. (Cloud vFW, Exchange, Local Network)	
			Dep. (Cloud DNS, Process, Remote Network)	
			Dep. (GCP Cloud Portal, Control, Local Network)	
App0 (SAP Application-1, Netweaver, 7.5)		Mgmt. ((443, GCP-Console), Agt0, Remote(Internet), GCP Admin group)	Dep. (SAP DB, (Process, Store), Local Network)	
			Dep. (Cloud Compute-1, Host, Direct)	
			Dep. (GCP Cloud Portal, Control, Local Network)	

**Table 6 sensors-22-05726-t006:** Data level and phase analysis.

Service (Actor, Asset, Data-Level)	Data Phases Information
Bs_0_ (USR, Agt_0_,bD)	(Dr)—Browser saves session information locally such as certificates, cookies, and history. Additionally, Data are stored in a single user profile for all employees. (Dp)—Non-business Open-source plugins are installed. (Dt)—Data are sent over wireless systems used for business and non-business purposes.
Bs_0_ (USR, Net_0,_bD)	(Dt)—PoS Modem is loaded with the default setting except for the wireless setting (WPA2 encryption). The admin page is available to access from any wireless SSID using HTTP as well as locally stored credentials.
Bs_0_ (USR, Net_3,_bD)	(Dt)—Cloud vFW external interface eth0 is configured with a public IP address using a basic setting. Data volume is not restricted. Traffic is allowed based on any source to the SAP app using the IP address and port (443). However, the admin console is not accessed from the internet.
Bs_0_ (USR, App_0,_bD)	(Dp)—SAP application is running as system privilege. The SAP application uses basic user names and passwords for user authentication. SAP applications exchange data during processing over multiple ports with SAP DB. (Dt)—SAP application data are sent over HTTPS to business users. SAP application data are forwarded to Net3 using private VN. SAP application authentication data are sent over LDAP protocol.
Bs_0_ (OPR, Agt_1,_mD)	(Dr)—GCP computes and cloud vFW identification information such as IP, port, and username is saved in the putty software for easy access. The private key file for cloud computing is stored locally on the system admin machine. (Dp)—The putty software is installed as a trusted system-level process. The putty software does not require authentication to run. (Dt)—The putty software sends data over ssh or sftp based on IP address as an identifier.

**Table 7 sensors-22-05726-t007:** Threat profile for Bs_0_.

Asset_id_ (Data-Level, Data-Phase)	Weaknesses	Threats	Criticality (Bt, Tc, Bi)
Agt0 ((bD, mD), (Dr, Dp, Dt))	CWE-494: Download of Code Without Integrity Check [35]	CAPEC-662: Adversary in the Browser (AiTB) [36]	(M, M, L) → L
Agt0 ((bD, mD), (Dr))	CWE-921: Storage of Sensitive Data in a Mechanism without Access Control [35]	CAPEC-196: Session Credential Falsification through Forging [36]	(L, L, M) → L
Agt1 (mD, Dr)	CWE-922: Insecure Storage of Sensitive Information [35]	CAPEC-529: Malware-Directed Internal Reconnaissance [36]	(M, M, H) → H
Net0 (md, Dt)	CWE-319: Cleartext Transmission of Sensitive Information [35]	CAPEC-102: Session Sidejacking [36]	(M, L, L) → VL
Net0 (mD, Dp)	CWE-284: Improper Access Control [35]	CAPEC-1: Accessing Functionality Not Properly Constrained by ACLs [36]	(M, M, L) → L
Net3 (mD, Dp)	CWE-308: Use of Single-factor Authentication [35]	CAPEC-151: Identity Spoofing [36]	(H, M, H) → VH
Net3 (bD, Dt)	CWE-770: Allocation of Resources Without Limits or Throttling [35]	CAPEC-125: Flooding [36]	(H, H, H) → VH
App0 ((bD, mD), Dp)	CWE-308: Use of Single-factor Authentication [35]	CAPEC-151: Identity Spoofing [36]	(H, M, H) → VH
App0 (bD, Dp)	CWE-20: Improper Input Validation [35]	CAPEC-63: Cross-Site Scripting (XSS) [37]	(H, H, H) → VH

**Table 8 sensors-22-05726-t008:** A table representing threat controls to protect (Bs_0_) data.

Asset_id_ (Data-Level, Data-Phase)	Threats (Criticality)	Controls	Assurance Level (Ct, Ef, Cx) → OAL
Net_3_ (mD, Dp)	CAPEC-151 (VH)	IA-2(1)-Identification Additionally, Authentication [38]	(H, H, M) → 8 (H)
Net_3_ (bD, Dt)	CAPEC-125 (VH)	SC-5(3)-Denial-Of-Service Protection [38]	(H, M, L) → 8(H)
App_0_ ((bD, mD), Dp)	CAPEC-151 (VH)	IA-2(1)-Identification Additionally, Authentication [38]	(H, H, M) → 8 (H)
App_0_ (bD, Dp)	CAPEC-63 (VH)	SI-10(5)-Information Input Validation [38]	(H, H, L) → 9 (H)

**Table 9 sensors-22-05726-t009:** d-TM Threat modelling comparison.

Factors\Models	PASTA	STRIDE	d-TM
Threat Modelling Methodology	Attack-Centric	Threat-centric	Data-Centric
Stages	7	N/A	4
Consideration of Business	Yes	No	Yes
Threats Criticality and Control	Yes	No	Yes
Identification of Attack Surface	Attacks to: - Network - Compute - Application	Threats to: - Compute - Application	Threat to Data in: - User agent - Network - Compute - Application - Storage
Data Area of Focus	Single category, three phases	Single category	Three categories: Management/Control/Business Three phases.
Threat Modelling Components	- Business - Asset/Application - Motivation/Scenario - Vulnerability - Control	- Asset - Vulnerability - Threat - Control	- Business/Asset - Data/Vulnerability - Threat - Control - Assurance

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
