# Peer review of "Data-Driven Threat Analysis for Ensuring Security in Cloud Enabled Systems"

_sensors, 2022, doi:10.3390/s22155726_

Round 1
Reviewer 1 Report
This paper proposes a threat analysis based on the d-TM approach to support identifying, analyzing, and controlling threats from three data abstraction levels and phases to secure the overall cloud-based environment.
There are some suggestions that authors must take into consideration to improve the quality of the paper:
- The abstract should be rewritten, highlighting the main contributions and emphasizing the motivation of this study.
- English must be improved in the document. There are some typos.
- The “Introduction” section must be rewritten to motivate the readers. It is necessary to present the problem, describe the section according to the proposed approach, and justify the importance of this issue.
- In the related work, it is recommended to refer to the contribution made by the researchers and the novelty of the research. However, the author does not mention that.
- I recommend that the authors add more current articles to improve the paper's overall quality. The preparation of a comparative analysis of the current publications on this subject should also be included.
- Some more clarification regarding the motivation and challenges of the proposed approach, and how the prescribed scheme would overcome them must be added in the new version.
- Improve the quality and resolution of figures 1, 2, 3, and 4 (at least 300 dpi) and explain those properly.
- Are there existing initiatives similar to those described in the conclusions and key findings?
- The research method should be better explained.
- This type of proposal should include figures that represent models or diagrams concerning the processes’ explanation. It is missing in the paper.
- Quantitative analysis or statistical analysis is required to compare this approach with others. How can you provide it?
- It is not clear the evidence of your approach implementation over Google Cloud (case study). What do you develop as a part of the approach? Algorithms, intelligent agents? - What programming languages were used to implement the approach? More details and descriptions are needed.
- Try to reduce the number of pages because it is hard to read the paper.
Author Response
Authors’ response to Reviewers’ Comments
Sensors (ISSN 1424-8220)MDPI
Manuscript ID sensors-1790103
Title: Data-driven Threat Analysis for Ensuring Security in Cloud enabled Systems
We want to thank the reviewers for their time and effort to review our paper and for their comments. The new version of the paper addresses all the comments raised by the reviewers. The text below indicates our responses to those comments and points out areas of the paper that have been modified based on the comments. We have added both track change and without track change version of the revised paper.
To facilitate a more straightforward reading of our responses, we have used the following format in the text below:
- Each comment has been highlighted in bold.
- After each reviewer’s comment, our corresponding reply is provided.
Reviewer 1 comments
Comments and Suggestions for Authors
Comment: This paper proposes a threat analysis based on the d-TM approach to support identifying, analyzing, and controlling threats from three data abstraction levels and phases to secure the overall cloud-based environment. There are some suggestions that authors must take into consideration to improve the quality of the paper:
Response: We thank the reviewer for their feedback and comments. All the raised suggestions are addresses accordingly.
Comment: The abstract should be rewritten, highlighting the main contributions and emphasizing the motivation of this study.
Response: The abstract is now revised by following this comment . The main contributions and motivation of the paper is clear now from the abstract.
Comment: English must be improved in the document. There are some typos.
Response: We have fully revised the paper for the grammar , typos, and correct sentence construction. We believe the current version complete and address all the English issues.
Comment: - The “Introduction” section must be rewritten to motivate the readers. It is necessary to present the problem, describe the section according to the proposed approach, and justify the importance of this issue.
Response: By following this comment, we have revised the introduction accordingly. The main limitation based on the existing contributions is the lack of focus on the comprehensive understanding of data for the threat analysis. The data migrates into the cloud service provider infrastructure incurs a number of challenges and it is necessary to address these challenges for a secure cloud base service delivery. This paper presents a novel threat modelling approach that classifies the data and gathers common weaknesses and threats so that appropriate control actions can be taken into consideration to tackle the threats. The approach is significantly important because it places a broad emphasis on data and its types for the threat analysis. The proposed approach classifies data in order to distinguish between the various risks associated with each type as well as which data are more susceptible to threat. This information is added in the introduction section.
Comment: - In the related work, it is recommended to refer to the contribution made by the researchers and the novelty of the research. However, the author does not mention that. I recommend that the authors add more current articles to improve the paper's overall quality. The preparation of a comparative analysis of the current publications on this subject should also be included. Some more clarification regarding the motivation and challenges of the proposed approach, and how the prescribed scheme would overcome them must be added in the new version.
Response: Thank you for this comment. By following this comment, we have added the most recent works related to the cloud threat domain. The section 2.1 reviews the existing literature in the threat analysis models and standards. Section 2.2 adds the works relating to the cloud threat analysis. Finally, a summary of the related work is added which indicates the limitations of the existing works . In particular, the majority of threat analysis is focused on topics like assets, strategies, and threat modelling. Furthermore, standards such as NIST SP 800-154 has concentrated on business data for systems that lack a thorough understanding of all data from the entire cloud system environment, such as control data. A comparison of the existing works and the result of our work is added in the discussion section 4.
The proposed d-TM aims to address the limitations of the existing works and contributes to improve the threat analysis practice by taking into account data and its level within overall cloud eco system and analyse and control the threats accordingly. Thus, our work makes four main contributions . Firstly, the proposed data-driven threat model (d-TM) describes the data abstraction in terms of three levels, i.e., control, management, business and phases. Each data level has its own set of security concerns, that needs adequate protection. Secondly, a systematic process is added to understand organisational data and services and its data flow for the threat analysis and controls. Thirdly, it adopts the common widely used security knowledge from existing standards including Common Attack Pattern Enumeration and Classification (CAPEC), Common Weakness Enumeration (CWE) and NIST SP800-53 for the threat analysis and control. Finally, a real cloud-based use case scenario is used to evaluate the applicability of the d-TM.
We have added text relating to the contribution of our work and motivation end of introduction and related work, discussion, and conclusion.
Comment- Improve the quality and resolution of figures 1, 2, 3, and 4 (at least 300 dpi) and explain those properly.
Response: By following this comment, resolution of the Figure 1,2,3 and 4 has been improved to 500dpi. The explanation about the figures is also added accordingly.
Comment- Are there existing initiatives similar to those described in the conclusions and key findings?
Response: There are a number of threat modelling approaches including STRIDE, PASTA and attack tress. Additionally, NIST SP 800-154 also provides data-centric system threat modelling to support the risk management activities. The standard considers attack and defense side information for data based on data location ,movement , objective and process. There are also initiatives that focus on analysing threats and risks for the cloud base service. An overview of these approached and key finding is added in the related work section. Moreover, a comparison of some of the existing works and results from the studied context by adopting proposed d-TM is also added in the discussion section.
Comment - The research method should be better explained.
Response: By following this comment, we have revised the research method. In particular, the research methodology used by d-TM is initiated considering a number of concepts including five different threat layers, actors and common knowledgebase. The methodology then considers threat analysis process based on the concepts and knowledgebase. Finally, the method is evaluated using a cloud based case study scenario and results from the studied context is compared with the existing works to generalised the findings. The text is added in Section 4.
Comment- This type of proposal should include figures that represent models or diagrams concerning the processes’ explanation. It is missing in the paper.
Response: Thanks for the comment. Figure 2 provides an overview of the proposed d-TM. The threat analysis process consists of four phases. The initial two phases give the necessary knowledge of the business processes, services, and assets so that DFD can be generated to demonstrate the data flow in various phases. The threat analysis phase follows, in which assets and identified data are evaluated for weaknesses and related threats. The threats are prioritized in order to determine their threat criticality to business data. Eventually, all information obtained from this step is merged into a threat profile table. The last phase attempts to offer appropriate controls to reduce prioritized threats. An overview of the process is added in section 4.3.
Comment- Quantitative analysis or statistical analysis is required to compare this approach with others. How can you provide it?
Response: The proposed d-TM is applied into a real cloud based use case context. We compared the findings from the studied context with the existing works.
Comment - It is not clear the evidence of your approach implementation over Google Cloud (case study). What do you develop as a part of the approach? Algorithms, intelligent agents? - What programming languages were used to implement the approach? More details and descriptions are needed.
Response: The case study uses S/4HANA application, which is hosted in GCP, the case study uses Infrastructure as a Service (IaaS) model. The evaluation is considering three assets that are hosted in cloud such as cloud gateway, cloud compute and cloud application. Our approach develops a data-driven threat analysis to identify , analyse, and control threats to secure cloud based system. A detailed description of the d-TM and evaluation of d-TM added in section 4 and 5 . Note that there is no programming language, algorithms or infrastructure is deployed.
Comment - Try to reduce the number of pages because it is hard to read the paper.
Response: We have revised the whole paper and reduced unnecessary text. Please note that there are comments by the reviewers, which need to add new text. We managed to reduce fives pages comparing to the first version.
Reviewer 2 Report
mistake
it is necessary to expand the information about related works (2) - missing section 2.2
to consider the levels, the OSI model should be used (section 3)
figure 3 should be described in more detail, or excluded
the experimental part should be described in more detail and compared with similar approaches
Author Response
Authors’ response to Reviewers’ Comments
Sensors (ISSN 1424-8220)MDPI
Manuscript ID sensors-1790103
Title: Data-driven Threat Analysis for Ensuring Security in Cloud enabled Systems
We want to thank the reviewers for their time and effort to review our paper and for their comments. The new version of the paper addresses all the comments raised by the reviewers. The text below indicates our responses to those comments and points out areas of the paper that have been modified based on the comments. We have added both track change and without track change version of the revised paper.
To facilitate a more straightforward reading of our responses, we have used the following format in the text below:
- Each comment has been highlighted in bold.
- After each reviewer’s comment, our corresponding reply is provided.
Reviewer 2 comments
Comments and Suggestions for Authors mistake
Comment : it is necessary to expand the information about related works (2) - missing section 2.2 to consider the levels, the OSI model should be used (section 3)
Response: By following this comment, we have extended the related work section 2. The section 2.1 focuses on threat analysis model and standards and section 2.2 focuses on the existing works relating to the cloud threat. A number of recent works are reviewed in section 2.2. The proposed approach considers five different threat layers in cloud computing including agent, network, compute , application and storage. The information is added in section 4.2. OSI model is not used by the proposed d-TM. The threat layered approach used is more relevant to data abstraction levels and cloud infrastructure.
Comment : figure 3 should be described in more detail, or excluded
Response: Figure 3 provides an overall architecture of the studied context. It includes both internal and outsourced infrastructure . In particular, sales operation is hosted to the cloud service. However, the remaining services, on the other hand, are located in the local data center. Therefore, the Figure 3 is necessary for the evaluation section. Detailed text is added to explain the Figure 3.
Comment : the experimental part should be described in more detail and compared with similar approaches
Response: By following this comment, we have revised the evaluation and discussion section. We have provided the detailed about the data collection from the studied context and identified possible threats based on the business processes and three different data levels. The threats are analysed and appropriate controls are identified to ensure security of cloud enabled system. We have also compared the findings from the studied context with the existing contribution in the discussion section. In particular, several threats are identified which are similar to the other findings. For instance, OWASP "Top 10 Cloud Risks", which highlights two issues revealed by the proposed threat analysis method, "User Identity Federation" and "Infrastructure Security". Furthermore, Denial-of-Service(DoS) attack, which is discovered also in our scenario and been recognized by other works However, the majority of the similarities in detected threats are related to cloud applications or systems. Moreover, the analysis also able to uncovered additional threats that not covered by others which is related to underlying infrastructure such as Agt0, Net0 and Net3 in our scenario. d-TM includes DFD approach with 9 weaknesses based on CWE, 9 threats described by CAPEC and 3 NIST controls. all are identified form the studied context. The proposed d-TM is unique due to the consideration of the abstraction of data levels and business processes and services for the threat analysis. A detailed about the comparison is added in the discussion section.
Reviewer 3 Report
This paper presents a data-driven approach for threat analysis, called d-TM, for cloud-based systems.
The paper sounds good and deals with a relevant problem...
However, far beyond the scientific merit of the proposed approach, the paper is too long.... so it is difficult to read...
The authors should focus on the innovative aspects of the proposed approach without getting lost beyond secondary aspects.... so the authors should avoid primarily explaining ideas that are well-known in the literature...
In fact, the paper has to be re-worked so as to make the research contributions characterizing the paper clearer...
Author Response
Authors’ response to Reviewers’ Comments
Sensors (ISSN 1424-8220)MDPI
Manuscript ID sensors-1790103
Title: Data-driven Threat Analysis for Ensuring Security in Cloud enabled Systems
We want to thank the reviewers for their time and effort to review our paper and for their comments. The new version of the paper addresses all the comments raised by the reviewers. The text below indicates our responses to those comments and points out areas of the paper that have been modified based on the comments. We have added both track change and without track change version of the revised paper.
To facilitate a more straightforward reading of our responses, we have used the following format in the text below:
- Each comment has been highlighted in bold.
- After each reviewer’s comment, our corresponding reply is provided.
Reviewer 3 comments
Comments and Suggestions for Authors
Comment : This paper presents a data-driven approach for threat analysis, called d-TM, for cloud-based systems. The paper sounds good and deals with a relevant problem.... However, far beyond the scientific merit of the proposed approach, the paper is too long.... so it is difficult to read.... The authors should focus on the innovative aspects of the proposed approach without getting lost beyond secondary aspects.... so the authors should avoid primarily explaining ideas that are well-known in the literature.... In fact, the paper has to be re-worked so as to make the research contributions characterizing the paper clearer...
Response: Thanks for this comment. We have reviewed the whole paper and reduced the secondary aspects. Please note that there are comments by the other reviewers which require to add additional text specifically in related work , evaluation and discussion section. But we managed to reduce five pages comparing to the initial version.
This paper presents a novel threat modelling approach that initiates from understanding business processes, services and infrastructure. Infrastructure is categorized to five attack layers considering data from user agent to data storage. The model classifies the data and uses DFD diagram , which aim to gathers common weaknesses and threats so that appropriate control actions can be taken into consideration to tackle the threats. The approach is significantly important because it places a broad emphasis on data and its types for the threat analysis. The d-TM classifies data in order to distinguish between the various risks associated with each type as well as which data are more susceptible to threat. There are four main contributions of this work. The text related to the main contribution is added in the introduction and discussion section.
Additionally, we have also compared the findings of the proposed approach with the existing contributions. In particular, several threats are identified which are similar to the other findings. For instance, OWASP "Top 10 Cloud Risks", which highlights two issues revealed by the proposed threat analysis method, "User Identity Federation" and "Infrastructure Security". Furthermore, Denial-of-Service(DoS) attack, which is discovered also in our scenario and been recognized by other works However, the majority of the similarities in detected threats are related to cloud applications or systems. Moreover, the analysis also able to uncovered additional threats that not covered by others which is related to underlying infrastructure.
The proposed d-TM presents a significant value toward existing threat models, that due to many features that empower the model. The model considers business processes and services as initial reference point to analyse threats, which is overlook by existing contributions. It also defined attacks layers and data levels for the threat analysis. In d-TM add another great value compared to others models that keeps attack surfaces to expert judgments, that could result inconsistency in reproducing results or overlooking a necessary attack surface. As a result of business processes, services and layered enabled infrastructure understanding, d-TM translate this information to a data-driven DFD diagram that present data types and phases any infrastructure asset. The proposed d-TM is unique due to the consideration of the abstraction of data levels and business processes and services for the threat analysis.
We have reworked the whole paper and merit of the proposed approach is added in the introduction, discussion, and conclusion section.
Reviewer 4 Report
The author focuses on the field of cloud computing and they introduce a data driven approach in order to analyze the threats that can occur in cloud based systems. The proposed threat analysis considers data that are collected from three different levels ranging from the management to the control and the business environment as well last three phases of storage, process, and transmission. The authors have provided a detailed qualitative analysis of the proposed model and some indicative qualitative results in order to discuss the benefits of the proposed framework. The provided analysis is concrete, complete, and correct and the authors have well thought out their main contributions. The structure of the paper is good enabling the reader to easily follow the provided analysis. The authors are encouraged to consider the following suggestions provided by the reviewer in order to improve the scientific depth of their manuscript, as well as they need to address the following comments in order to improve the quality of presentation of their manuscript. Initially, the provided discussion of the related work presented in sections one and two needs to be substantially revised in order to be presented by using more summative language towards enabling the average reader to follow the main research contributions that have already been performed in the literature and the research gaps that the authors try to address. In Section 2, the authors need to discuss existing approaches that exploit the software defined technology, such as Mitsis, G., et al. "Intelligent dynamic data offloading in a competitive mobile edge computing market." Future Internet 11.5 (2019): 118, in order to deal with the advanced computing demand in next generation computing systems. In section 3, the authors need to discuss how they will collect data related to the management, the control, the business level in a realistic implementation of the proposed novel framework. In Section 4, the authors need to include an additional subsection discussing the implementation cost of the proposed analysis. In section 5, the authors need to provide at least a qualitative comparison to the state of the art in order to identify the drawbacks and benefits of the proposed approach. Finally, the overall manuscript needs to be checked for typos, syntax, and grammar errors in order to improve the quality of its presentation.
Author Response
Authors’ response to Reviewers’ Comments
Sensors (ISSN 1424-8220)MDPI
Manuscript ID sensors-1790103
Title: Data-driven Threat Analysis for Ensuring Security in Cloud enabled Systems
We want to thank the reviewers for their time and effort to review our paper and for their comments. The new version of the paper addresses all the comments raised by the reviewers. The text below indicates our responses to those comments and points out areas of the paper that have been modified based on the comments. We have added both track change and without track change version of the revised paper.
To facilitate a more straightforward reading of our responses, we have used the following format in the text below:
- Each comment has been highlighted in bold.
- After each reviewer’s comment, our corresponding reply is provided.
Reviewer 4 comments
Comments and Suggestions for Authors
Comment: The author focuses on the field of cloud computing and they introduce a data driven approach in order to analyze the threats that can occur in cloud based systems. The proposed threat analysis considers data that are collected from three different levels ranging from the management to the control and the business environment as well last three phases of storage, process, and transmission. The authors have provided a detailed qualitative analysis of the proposed model and some indicative qualitative results in order to discuss the benefits of the proposed framework. The provided analysis is concrete, complete, and correct and the authors have well thought out their main contributions. The structure of the paper is good enabling the reader to easily follow the provided analysis. The authors are encouraged to consider the following suggestions provided by the reviewer in order to improve the scientific depth of their manuscript, as well as they need to address the following comments in order to improve the quality of presentation of their manuscript.
Response: Thank you for this comment. We have addressed all the comments and detailed is given below.
Comment: Initially, the provided discussion of the related work presented in sections one and two needs to be substantially revised in order to be presented by using more summative language towards enabling the average reader to follow the main research contributions that have already been performed in the literature and the research gaps that the authors try to address.
Response: By following this comment, we have revised the whole related work section. The new section 2.1 focuses on the existing work relating to threat modelling and standard and section 2.2 added new works relating to the threat and risks management for cloud computing. In particular, a number of existing contributions identify the key threats in cloud including data manipulation and leaking. Other works focus on the threat intelligence parameters such as Matrix IoC, attack , behavior and pattern are used for the threat analysis.
To summarize, the works reviewed above provide feasible approaches to understanding and analyzing cloud risks in domains where traditional threat modelling methodology does not place a strong emphasis on data. The majority of threat analysis is focused on topics like assets, strategies, and threat modelling. Furthermore, standards such as NIST SP 800-154 has concentrated on business data for systems that lack a thorough understanding of all data from the entire cloud system environment, such as control data. Our study solves these issues by providing a native data-driven threat analysis model that incorporates a full technological coverage of numerous types of data in the cloud at any point of its lifetime. The proposed approach also considers business processes and services as initial reference point to analyse threats, which is overlook by existing contributions.
This information is added in the related work section.
Comment: In Section 2, the authors need to discuss existing approaches that exploit the software defined technology, such as Mitsis, G., et al. "Intelligent dynamic data offloading in a competitive mobile edge computing market." Future Internet 11.5 (2019): 118, in order to deal with the advanced computing demand in next generation computing systems.
Response: Thank you for referring this interesting work. We have reviewed this work and added in section 2.2.
Comment : In section 3, the authors need to discuss how they will collect data related to the management, the control, the business level in a realistic implementation of the proposed novel framework. In Section 4, the authors need to include an additional subsection discussing the implementation cost of the proposed analysis.
Response: By following this comment, we have added text in section 3, 4 and 5 . Section 3 presents the data abstraction from management, control and business level . Examples are added for these levels which indicate possible data collection for each level. Additionally, the detailed about the data collection and analysis are elaborated in Phase 1 and 2 of section 4.3. The implementation of the proposed approach is mainly discussed in evaluation section 5. Please note that the focus of the evaluation is mainly determining the applicability of the proposed approach in terms of threat identification and control for ensuing security of the cloud based system.
Comment: In section 5, the authors need to provide at least a qualitative comparison to the state of the art in order to identify the drawbacks and benefits of the proposed approach.
Response: By following this comment, we have compared the findings from the evaluation with the existing contributions. . In particular, several threats are identified which are similar to the other findings. For instance, OWASP "Top 10 Cloud Risks", which highlights two issues revealed by the proposed threat analysis method, "User Identity Federation" and "Infrastructure Security". Furthermore, Denial-of-Service(DoS) attack, which is discovered also in our scenario and been recognized by other works However, the majority of the similarities in detected threats are related to cloud applications or systems. Moreover, the analysis also able to uncovered additional threats that not covered by others which is related to underlying infrastructure.
The proposed d-TM presents a significant value toward existing threat models, that due to many features that empower the model. The model considers business processes and services as initial reference point to analyse threats, which is overlook by existing contributions. It also defined attacks layers and data levels for the threat analysis. In d-TM add another great value compared to others models that keeps attack surfaces to expert judgments, that could result inconsistency in reproducing results or overlooking a necessary attack surface. As a result of business processes, services and layered enabled infrastructure understanding, d-TM translate this information to a data-driven DFD diagram that present data types and phases any infrastructure asset. The proposed d-TM is unique due to the consideration of the abstraction of data levels and business processes and services for the threat analysis.
The benefits and drawbacks of the proposed approach is added in the discussion section.
Comment : Finally, the overall manuscript needs to be checked for typos, syntax, and grammar errors in order to improve the quality of its presentation.
Response: The whole paper is now revised to improve the English
Round 2
Reviewer 1 Report
Although the comments and suggestions were taken into account, it is necessary to consider the following changes to improve the quality of the paper:
- There are some typos and wordy sentences... So, improve English, for instance, words such as "organization" are repeated several times in the same paragraphs. Like this example, there are other paragraphs with the same problem.
- In your answer, you mentioned that figures now have a resolution of 500 dpi, but they still look blurry or pixelated when you zoom in to more detail. Please review this issue.
- I reviewed the point: "Quantitative analysis or statistics is required to compare this approach with others. How can you provide it?"
This last comment is important because you can express the analysis with a table that contains a statistical analysis or the performance between approaches.
Author Response
Authors’ response to Reviewers’ Comments
Sensors (ISSN 1424-8220)MDPI
Manuscript ID sensors-1790103
Title: Data-driven Threat Analysis for Ensuring Security in Cloud enabled Systems
We want to thank the reviewers for their time and effort to review our paper and for their comments. The new version of the paper addresses all the comments raised by the reviewers. The text below indicates our responses to those comments and points out areas of the paper that have been modified based on the comments. We have added both track change and without track change version of the revised paper.
To facilitate a more straightforward reading of our responses, we have used the following format in the text below:
- Each comment has been highlighted in bold.
- After each reviewer’s comment, our corresponding reply is provided.
Reviewer 1 comments
Comment -Although the comments and suggestions were taken into account, it is necessary to consider the following changes to improve the quality of the paper:
Response: We thank the reviewer for their feedback and comments. All the raised suggestions are addresses accordingly.
Comment - There are some typos and wordy sentences... So, improve English, for instance, words such as "organization" are repeated several times in the same paragraphs. Like this example, there are other paragraphs with the same problem. In your answer, you mentioned that figures now have a resolution of 500 dpi, but they still look blurry or pixelated when you zoom in to more detail. Please review this issue.
Response: By following this comment, we have further revised the paper for typos and wordy sentence. We have also deleted repeated words such organisations in the same para.
We have also carefully looked at the figures and assure it quality is enhanced. The figures now look clear and not blurry. If still the issue exists, then this could be happened during the file upload.
Comment - I reviewed the point: "Quantitative analysis or statistics is required to compare this approach with others. How can you provide it?" This last comment is important because you can express the analysis with a table that contains a statistical analysis or the performance between approaches.
Response: We have compared the finding from the studied context with the existing work in the literature for the purpose of generalisation. Please note that the existing threat modelling approaches are not data focused and there is lack of data available relating to the implementation of the approaches. However, in response to this comment, we have added a table to compare the features of d-TM with the existing threat models including STRID, and PASTA.
Reviewer 2 Report
enlarge the pictures and make them more presentable
other comments are almost completely taken into account
Author Response
Authors’ response to Reviewers’ Comments
Sensors (ISSN 1424-8220)MDPI
Manuscript ID sensors-1790103
Title: Data-driven Threat Analysis for Ensuring Security in Cloud enabled Systems
We want to thank the reviewers for their time and effort to review our paper and for their comments. The new version of the paper addresses all the comments raised by the reviewers. The text below indicates our responses to those comments and points out areas of the paper that have been modified based on the comments. We have added both track change and without track change version of the revised paper.
To facilitate a more straightforward reading of our responses, we have used the following format in the text below:
- Each comment has been highlighted in bold.
- After each reviewer’s comment, our corresponding reply is provided.
Reviewer 2 comments
Comments and Suggestions for Authors mistake
Comment : enlarge the pictures and make them more presentable and other comments are almost completely taken into account
Response: Thank you for this comment. We have enlarged all figures.
Reviewer 3 Report
The paper can be accepted....
Author Response
Authors’ response to Reviewers’ Comments
Sensors (ISSN 1424-8220)MDPI
Manuscript ID sensors-1790103
Title: Data-driven Threat Analysis for Ensuring Security in Cloud enabled Systems
We want to thank the reviewers for their time and effort to review our paper and for their comments. The new version of the paper addresses all the comments raised by the reviewers. The text below indicates our responses to those comments and points out areas of the paper that have been modified based on the comments. We have added both track change and without track change version of the revised paper.
To facilitate a more straightforward reading of our responses, we have used the following format in the text below:
- Each comment has been highlighted in bold.
- After each reviewer’s comment, our corresponding reply is provided.
Reviewer 3 comments
Comments and Suggestions for Authors
Comment : The paper can be accepted....
Response: Thanks for accepting our paper.
Reviewer 4 Report
The comments of the reviewers have been addressed in detail by the authors. The authors have substantially improved the quality of presentation and the structure of the paper. The manuscript can be accepted in its current form.
Author Response
Authors’ response to Reviewers’ Comments
Sensors (ISSN 1424-8220)MDPI
Manuscript ID sensors-1790103
Title: Data-driven Threat Analysis for Ensuring Security in Cloud enabled Systems
We want to thank the reviewers for their time and effort to review our paper and for their comments. The new version of the paper addresses all the comments raised by the reviewers. The text below indicates our responses to those comments and points out areas of the paper that have been modified based on the comments. We have added both track change and without track change version of the revised paper.
To facilitate a more straightforward reading of our responses, we have used the following format in the text below:
- Each comment has been highlighted in bold.
- After each reviewer’s comment, our corresponding reply is provided.
Reviewer 4 comments
Comments and Suggestions for Authors
Comment: The comments of the reviewers have been addressed in detail by the authors. The authors have substantially improved the quality of presentation and the structure of the paper. The manuscript can be accepted in its current form.
Response: Thank you for this very positive feedback and accepting our paper.